# Analysis of Robust PCA via Local Incoherence

**Huishuai Zhang**
Department of EECS
Syracuse University
Syracuse, NY 13244
hzhan23@syr.edu

**Yi Zhou**
Department of EECS
Syracuse University
Syracuse, NY 13244
yzhou35@syr.edu

**Yingbin Liang**
Department of EECS
Syracuse University
Syracuse, NY 13244
yliang06@syr.edu

## Abstract

We investigate the robust PCA problem of decomposing an observed matrix into the sum of a low-rank and a sparse error matrices via convex programming Principal Component Pursuit (PCP). In contrast to previous studies that assume the support of the error matrix is generated by uniform Bernoulli sampling, we allow non-uniform sampling, i.e., entries of the low-rank matrix are corrupted by errors with unequal probabilities. We characterize conditions on error corruption of each individual entry based on the local incoherence of the low-rank matrix, under which correct matrix decomposition by PCP is guaranteed. Such a refined analysis of robust PCA captures how robust each entry of the low rank matrix combats error corruption. In order to deal with non-uniform error corruption, our technical proof introduces a new weighted norm and develops/exploits the concentration properties that such a norm satisfies.

## 1 Introduction

We consider the problem of robust Principal Component Analysis (PCA). Suppose a $n$-by-$n$[1] data matrix $M$ can be decomposed into a low-rank matrix $L$ and a sparse matrix $S$ as

$$M = L + S. \tag{1}$$

Robust PCA aims to find $L$ and $S$ with $M$ given. This problem has been extensively studied recently. In [1, 2], *Principal Component Pursuit (PCP)* has been proposed to solve the robust PCA problem via the following convex programming

$$\text{PCP:} \quad \underset{L,S}{\text{minimize}} \quad \|L\|_* + \lambda \|S\|_1 \tag{2}$$
$$\text{subject to} \quad M = L + S,$$

where $\| \cdot \|_*$ denotes the nuclear norm, i.e., the sum of singular values, and $\| \cdot \|_1$ denotes the $l_1$ norm i.e., the sum of absolute values of all entries. It was shown in [1, 2] that PCP successfully recovers $L$ and $S$ if the two matrices are distinguishable from each other in properties, i.e., $L$ is not sparse and $S$ is not low-rank. One important quantity that determines similarity of $L$ to a sparse matrix is the *incoherence* of $L$, which measures how column and row spaces of $L$ are aligned with canonical basis and between themselves. Namely, suppose that $L$ is a rank-$r$ matrix with SVD $L = U\Sigma V^*$, where $\Sigma$ is a $r \times r$ diagonal matrix with singular values as its diagonal entries, $U$ is a $n \times r$ matrix with columns as the left singular vectors of $L$, $V$ is a $n \times r$ matrix with columns as the right singular vectors of $L$, and $V^*$ denotes the transpose of $V$. The *incoherence* of $L$ is measured

by $\mu = \max\{\mu_0, \mu_1\}$, where $\mu_0$ and $\mu_1$ are defined as

$$\|U^*e_i\| \leq \sqrt{\frac{\mu_0 r}{n}}, \qquad \|V^*e_j\| \leq \sqrt{\frac{\mu_0 r}{n}}, \quad \text{for all } i, j = 1, \cdots, n \tag{3}$$

$$\|UV^*\|_\infty \leq \sqrt{\frac{\mu_1 r}{n^2}}. \tag{4}$$

Previous studies suggest that the incoherence crucially determines conditions on sparsity of $S$ in order for PCP to succeed. For example, Theorem 2 in [3] explicitly shows that the matrix $L$ with larger $\mu$ can tolerate only smaller error density to guarantee correct matrix decomposition by PCP. In all previous work on robust PCA, the incoherence is defined to be the maximum over all column and row spaces of $L$ as in (3) and (4), which can be viewed as the *global* parameter for the entire matrix $L$, and consequently, characterization of error density is based on such global (and in fact *the worst case*) incoherence.

In fact, each $(i, j)$ entry of the low rank matrix $L$ can be associated with a *local* incoherence parameter $\mu_{ij}$, which is less than or equal to the *global* parameter $\mu$, and then the allowable entry-wise error density can be potentially higher than that characterized based on the global incoherence. Thus, the total number of errors that the matrix can tolerate in robust PCA can be much higher than that characterized based on the global incoherence when errors are distributed accordingly. Motivated by such an observation, this paper aims to characterize conditions on error corruption of each entry of the low rank matrix based on the corresponding local incoherence parameter, which guarantee success of PCP. Such conditions imply how robust each individual entry of $L$ to resist error corruption. Naturally, the error corruption probability is allowed to be *non-uniform* over the matrix (i.e., locations of non-zero entries in $S$ are sampled non-uniformly).

We note that the notion of local incoherence was first introduced in [4] for studying the matrix completion problem, in which local incoherence determines the local sampling density in order to guarantee correct matrix completion. Here, local incoherence plays a similar role, and determines the maximum allowable error density at each entry to guarantee correct matrix decomposition. The difference lies in that local incoherence here depends on both localized $\mu_0$ and $\mu_1$ rather than only on localized $\mu_0$ in matrix completion due to further difficulty of robust PCA, in which locations of error corrupted entries are unknown, as pointed out in [1, 3].

**Our Contribution.** In this paper, we investigate a more general robust PCA problem, in which entries of the low rank matrix are corrupted by non-uniformly distributed Bernoulli errors. We characterize the conditions that guarantee correct matrix decomposition by PCP. Our result identifies the local incoherence (defined by localized $\mu_0$ and $\mu_1$ for each entry of the low rank matrix) to determine the condition that each local Bernoulli error corruption parameter should satisfy. Our results provide the following useful understanding of the robust PCA problem:

- Our characterization provides a localized (and hence more refined) view of robust PCA, and determines how robust each entry of the low rank matrix combats error corruption.
- Our results suggest that the total number of errors that the low-rank matrix can tolerate depends on how errors are distributed over the matrix.
- Via cluster problems, our results provide an evidence that $\mu_1$ is necessary in characterizing conditions for robust PCA.

In order to deal with non-uniform error corruption, our technical proof introduces a new weighted norm denoted by $l_{w(\infty)}$, which involves the information of both localized $\mu_0$ and $\mu_1$ and is hence different from the weighted norms introduced in [4] for matrix completion. Thus, our proof necessarily involves new technical developments associated with such a new norm.

**Related Work.** A closely related but different problem from robust PCA is *matrix completion*, in which a low-rank matrix is partially observed and is to be completed. Such a problem has been previously studied in [5–8], and it was shown that a rank-$r$ $n$-by-$n$ matrix can be provably recoverable by convex optimization with as few as $\Theta(\max\{\mu_0, \mu_1\}nr \log^2 n)^2$ observed entries. Later on, it was shown in [4] that $\mu_1$ does not affect sample complexity for matrix completion and hence $\Theta(\mu_0 nr \log^2 n)$ observed entries are sufficient for guaranteeing correct matrix completion. It was further shown in [9] that a coherent low-rank matrix (i.e., with large $\mu_0$) can be recovered with

$\Theta(nr \log^2 n)$ observations as long as the sampling probability is proportional to the leverage score (i.e., localized $\mu_0$). Our problem can be viewed as its counterpart in robust PCA, where the difference lies in the local incoherence in our problem depends on both localized $\mu_0$ and $\mu_1$.

Robust PCA aims to decompose an observed matrix into the sum of a low-rank matrix and a sparse matrix. In [2, 10], robust PCA with fixed error matrix was studied, and it was shown that the maximum number of errors in any row or column should be bounded from above in order to guarantee correct decomposition by PCP. Robust PCA with random error matrix was investigated in a number of studies. It has been shown in [1] that such decomposition can be exact with high probability if the percentage of corrupted entries is small enough, under the assumptions that the low-rank matrix is incoherent and the support set of the sparse matrix is uniformly distributed. It was further shown in [11] that if signs of nonzero entries in the sparse matrix are randomly chosen, then an adjusted convex optimization can produce exact decomposition even when the percentage of corrupted entries goes to one (i.e., error is dense). The problem was further studied in [1, 3, 12] for the case with the error-corrupted low-rank matrix only partially observed. Our work provides a more refined (i.e. entry-wise) view of robust PCA with random error matrix, aiming at understanding how local incoherence affects susceptibility of each matrix entry to error corruption.

## 2  Model and Main Result

### 2.1  Problem Statement

We consider the robust PCA problem introduced in Section 1. Namely, suppose an $n$-by-$n$ matrix $M$ can be decomposed into two parts: $M = L + S$, where $L$ is a low rank matrix and $S$ is a sparse (error) matrix. We assume that the rank of $L$ is $r$, and the support of $S$ is selected randomly but *non-uniformly*. More specifically, let $\Omega$ denote the support of $S$ and then $\Omega \subseteq [n] \times [n]$, where $[n]$ denotes the set $\{1, 2, \ldots, n\}$. The event $\{(i, j) \in \Omega\}$ is independent across different pairs $(i, j)$ and

$$\mathbb{P}\left((i, j) \in \Omega\right) = \rho_{ij}, \tag{5}$$

where $\rho_{ij}$ represents the probability that the $(i, j)$-entry of $L$ is corrupted by error. Hence, $\Omega$ is determined by Bernoulli sampling with non-uniform probabilities.

We study both the *random sign* and *fixed sign* models for $S$. For the fixed sign model, we assume signs of nonzero entries in $S$ are arbitrary and fixed, whereas for the random sign model, we assume that signs of nonzero entries in $S$ are independently distributed Bernoulli variables, randomly taking values $+1$ or $-1$ with probability $1/2$ as follows:

$$[\operatorname{sgn}(S)]_{ij} = \begin{cases} 1 & \text{with prob.} \quad \rho_{ij}/2 \\ 0 & \text{with prob.} \quad 1 - \rho_{ij} \\ -1 & \text{with prob.} \quad \rho_{ij}/2. \end{cases} \tag{6}$$

In this paper, our goal is to characterize conditions on $\rho_{ij}$ that guarantees correct recovery of $L$ and $S$ with observation of $M$.

We provide some notations that are used throughout this paper. A matrix $X$ is associated with five norms: $\|X\|_F$ denotes the Frobenius norm, $\|X\|_*$ denotes the nuclear norm (i.e., the sum of singular values), $\|X\|$ denotes the spectral norm (i.e., the largest singular value), and $\|X\|_1$ and $\|X\|_\infty$ represent respectively the $l_1$ and $l_\infty$ norms of the long vector stacked by $X$. The inner product between two matrices is defined as $\langle X, Y \rangle := trace(X^*Y)$. For a linear operator $\mathcal{A}$ that acts on the space of matrices, $\|\mathcal{A}\|$ denotes the operator norm given by $\|\mathcal{A}\| = \sup_{\{\|X\|_F = 1\}} \|\mathcal{A}X\|_F$.

### 2.2  Main Theorems

We adopt the PCP to solve the robust PCA problem. We define the following *local* incoherence parameters, which play an important role in our characterization of conditions on entry-wise $\rho_{ij}$.

$$\mu_{0ij} := \frac{n}{2r}\left(\|U^*e_i\|^2 + \|V^*e_j\|^2\right), \quad \mu_{1ij} := \frac{n^2([UV^*]_{ij})^2}{r} \tag{7}$$

$$\mu_{ij} := \max\{\mu_{0ij}, \mu_{1ij}\}. \tag{8}$$

It is clear that $\mu_{0ij} \leq \mu_0$ and $\mu_{1ij} \leq \mu_1$ for all $i, j = 1, \cdots, n$. We note that although $\max_{i,j} \mu_{ij} > 1$, some $\mu_{ij}$ might take values as small as zero.

We first consider the robust PCA problem under the *random sign model* as introduced in Section 2.1. The following theorem characterizes the condition that guarantees correct recovery by PCP.

**Theorem 1.** *Consider the robust PCA problem under the random sign model. If*

$$1 - \rho_{ij} \geq \max\left\{ C_0 \sqrt{\frac{\mu_{ij} r}{n}} \log n, \frac{1}{n^3} \right\}$$

*for some sufficiently large constant $C_0$ and for all $i, j \in [n]$, then $PCP$ yields correct matrix recovery with $\lambda = \frac{1}{32\sqrt{n \log n}}$, with probability at least $1 - cn^{-10}$ for some constant c.*

We note that the term $1/n^3$ is introduced to justify dual certificate conditions in the proof (see Appendix A.2). We further note that satisfying the condition in Theorem 1 implies $C_0 \sqrt{\mu r/n} \log n \leq 1$, which is an essential bound required in our proof and coincides with the conditions in previous studies [1, 12]. Although we set $\lambda = \frac{1}{32\sqrt{n \log n}}$ for the sake of proof, in practice $\lambda$ is often determined via cross validation.

The above theorem suggests that the local incoherence parameter $\mu_{ij}$ is closely related to how robust each entry of $L$ to error corruption in matrix recovery. An entry corresponding to smaller $\mu_{ij}$ tolerates larger error density $\rho_{ij}$. This is consistent with the result in [4] for matrix completion, in which smaller local incoherence parameter requires lower local sampling rate. The difference lies in that here both $\mu_{0ij}$ and $\mu_{1ij}$ play roles in $\mu_{ij}$ whereas only $\mu_{0ij}$ matters in matrix completion. The necessity of $\mu_{1ij}$ for robust PCA is further demonstrated in Section 2.3 via an example.

Theorem 1 also provides a more refined view for robust PCA in the dense error regime, in which the error corruption probability approaches one. Such an interesting regime was previously studied in [3, 11]. In [11], it is argued that PCP with adaptive $\lambda$ yields exact recovery even when the error corruption probability approaches one if errors take random signs and the dimension $n$ is sufficiently large. In [3], it is further shown that PCP with a fixed $\lambda$ also yields exact recovery and the scaling behavior of the error corruption probability is characterized. The above Theorem 1 further provides the scaling behavior of the *local entry-wise* error corruption probability $\rho_{ij}$ as it approaches one, and captures how such scaling behavior depends on local incoherence parameters $\mu_{ij}$. Such a result implies that robustness of PCP depends not only on the error density but also on how errors are distributed over the matrix with regard to $\mu_{ij}$.

We next consider the robust PCA problem under the *fixed sign model* as introduced in Section 2.1. In this case, non-zero entries of the error matrix $S$ can take arbitrary and fixed values, and only locations of non-zero entries are random.

**Theorem 2.** *Consider the robust PCA problem under the fixed sign model. If*

$$(1 - 2\rho_{ij}) \geq \max\left\{ C_0 \sqrt{\frac{\mu_{ij} r}{n}} \log n, \frac{1}{n^3} \right\}$$

*for some sufficient large constant $C_0$ and for all $i, j \in [n]$, then $PCP$ yields correct recovery with $\lambda = \frac{1}{32\sqrt{n \log n}}$, with probability at least $1 - cn^{-10}$ for some constant c.*

Theorem 2 follows from Theorem 1 by adapting the elimination and derandomization arguments [1, Section 2.2] as follows. Let $\boldsymbol{\rho}$ be the matrix with each $(i, j)$-entry being $\rho_{ij}$. If PCP yields exact recovery with a certain probability for the random sign model with the parameter $2\boldsymbol{\rho}$, then it also yields exact recovery with at least the same probability for the fixed sign model with locations of non-zero entries sampled using Bernoulli model with the parameter $\boldsymbol{\rho}$.

We now compare Theorem 2 for robust PCA with *non-uniform* error corruption to Theorem 1.1 in [1] for robust PCA with *uniform* error corruption. It is clear that if we set $\rho_{i,j} = \rho$ for all $i, j \in [n]$, then the two models are the same. It can then be easily checked that conditions $\sqrt{\mu r/n} \log n \leq \rho_r$ and $\rho \leq \rho_s$ in Theorem 1.1 of [1] implies the conditions in Theorem 2. Thus, Theorem 2 provides a more relaxed condition than Theorem 1.1 in [1]. Such benefit of condition relaxation should be attributed to the new golfing scheme introduced in [3, 12], and this paper provides a more refined view of robust PCA by further taking advantage of such a new golfing scheme to analyze local conditions.

More importantly, Theorem 2 characterizes relationship between local incoherence parameters and local error corruption probabilities, which implies that different areas of the low-rank matrix have

different levels of ability to resist errors: a more incoherent area (i.e., with smaller $\mu_{ij}$) can tolerate more errors. Thus, Theorem 2 illustrates the following interesting fact. Whether PCP yields correct recovery depends not only on the total number of errors but also on how errors are distributed. If more errors are distributed to more incoherent areas (i.e, with smaller $\mu_{ij}$), then more errors in total can be tolerated. However, if errors are distributed in an opposite manner, then only smaller number of errors can be tolerated.

## 2.3 Implication on Cluster Matrix

In this subsection, we further illustrate our result when the low rank matrix is a cluster matrix. Although robust PCA and even more sophisticated approaches have been applied to solve clustering problems, e.g., [13–15], our perspective here is to demonstrate how local incoherence affects entry-wise robustness to error corruption, which has not been illustrated in previous studies.

Suppose there are $n$ elements to be clustered. We use a cluster matrix $L$ to represent the clustering relationship of these $n$ elements with $L_{ij} = 1$ if elements $i$ and $j$ are in the same cluster and $L_{ij} = 0$ otherwise. Thus, with appropriate ordering of the elements, $L$ is a block diagonal matrix with all diagonal blocks containing all '1's and off-diagonal blocks containing all '0's. Hence, the rank $r$ of $L$ equals the number of clusters, which is typically small compared to $n$. Suppose these entries are corrupted by errors that flip entries from one to zero or from zero to one. This can be thought of as adding a (possibly sparse) error matrix $S$ to $L$ so that the observed matrix is $L + S$. Then PCP can be applied to recover the cluster matrix $L$.

We first consider an example with clusters having equal size $n/r$. We set $n = 600$ and $r = 4$ (i.e., four equal-size clusters). We apply errors to diagonal-block entries and off-diagonal-block entries respectively with the probabilities $\rho_d$ and $\rho_{od}$. In Fig. 1a, we plot recovery accuracy of PCP for each pairs of $(\rho_{od}, \rho_d)$. It is clear from the figure that failure occurs for larger $\rho_{od}$ than $\rho_d$, which thus implies that off-diagonal blocks are more robust to errors than diagonal blocks. This can be explained by Theorem 2 as follows. For a cluster matrix with equal cluster size $n/r$, the local incoherence parameters are given by

$$\mu_{0ij} = 1 \text{ for all } (i,j), \quad \text{and} \quad \mu_{1ij} = \begin{cases} r, & (i,j) \text{ is in diagonal blocks} \\ 0, & (i,j) \text{ is in off-diagonal blocks,} \end{cases}$$

and thus

$$\mu_{ij} = \max\{\mu_{0ij}, \mu_{1ij}\} = \begin{cases} r, & (i,j) \text{ is in diagonal blocks} \\ 1, & (i,j) \text{ is in off-diagonal blocks.} \end{cases}$$

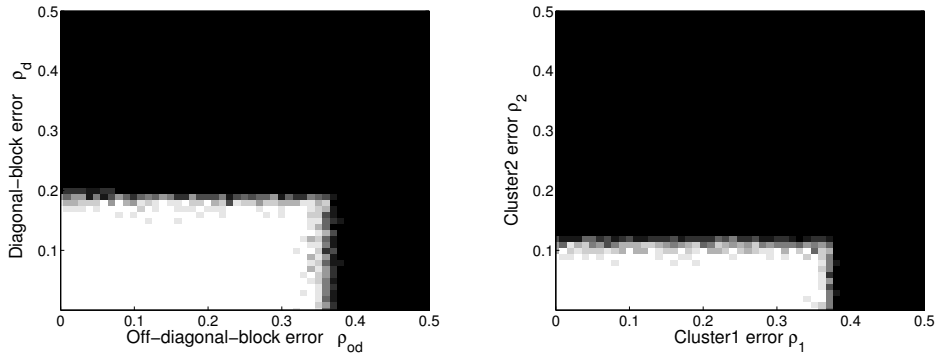

(a) Diagonal-block error vs. off-diagonal-block error. $n = 600, r = 4$ with equal cluster sizes

(b) Error vulnerability with respect to cluster sizes 500 vs. 100

Figure 1: Error vulnerability on different parts for cluster matrix. In both cases, for each probability pair, we generate 10 trials of independent random error matrices and count the number of successes of PCP. We declare a trial to be successful if the recovered $\hat{L}$ satisfies $\|\hat{L} - L\|_F / \|L\|_F \leq 10^{-3}$. Color from white to black represents the number of successful trials changes from 10 to 0.

Based on Theorem 2, it is clear that diagonal-block entries are more locally coherent and hence are more vulnerable to errors, whereas off-diagonal-block entries are more locally incoherent and hence are more robust to errors.

Moreover, this example also demonstrates the necessity of $\mu_1$ in the robust PCA problem. [4] showed that $\mu_1$ is not necessary for matrix completion and argued informally that $\mu_1$ is necessary for robust PCA by connecting the robust PCA problem to hardness of finding a small clique in a large random graph. Here, the above example provides an evidence for such a fact. In the example, $\mu_{0ij}$ are the same over the entire matrix, and hence it is $\mu_{1ij}$ that differentiates incoherence between diagonal blocks and off-diagonal blocks, and thus differentiates their robustness to errors.

We then consider the case with two clusters that have different sizes: cluster1 size 500 versus cluster2 size 100. Hence, $r = 2$. We apply errors to block diagonal entries corresponding to clusters 1 and 2 respectively with the probabilities $\rho_1$ and $\rho_2$. In Fig. 1b, we plot the recovery accuracy of PCP for each pair of $(\rho_1, \rho_2)$. It is clear from the figure that failure occurs for larger $\rho_1$ than $\rho_2$, which thus implies that entries corresponding to the larger cluster are more robust to errors than entries corresponding to smaller clusters. This can be explained by Theorem 2 because the local incoherence of a block diagonal entry is given by $\mu_{ij} = \frac{n^2}{rK^2}$, where $K$ is the corresponding cluster size, and hence the error corruption probability should satisfy $1 - 2\rho_{ij} > C_0 \frac{\sqrt{n}}{K} \log n$ for correct recovery. Thus, a larger cluster can resist denser errors. This also coincides with the results on graph clustering in [13, 16].

## 2.4 Outline of the Proof of Theorem 1

The proof of Theorem 1 follows the idea established in [1] and further developed in [3, 12]. Our main technical development lies in analysis of non-uniform error corruption based on local incoherence parameters, for which we introduce a new weighted norm $l_{w(\infty)}$, and establish concentration properties and bounds associated with this norm. As a generalization of matrix infinity norm, $l_{w(\infty)}$ incorporates both $\mu_{0ij}$ and $\mu_{1ij}$, and is hence different from the weighted norms $l_{\mu(\infty)}$ and $l_{\mu(\infty,2)}$ in [9] by its role in the analysis for the robust PCA problem. We next outline the proof here and the detailed proofs are provided in Appendix A.

We first introduce some notations. We define the subspace $T := \{UX^* + YV^* : X, Y \in \mathbb{R}^{n \times r}\}$, where $U, V$ are left and right singular matrix of $L$. Then $T$ induces a projection operator $\mathcal{P}_T$ given by $\mathcal{P}_T(M) = UU^*M + MVV^* - UU^*MVV^*$. Moreover, $T^\perp$, the complement subspace to $T$, induces an orthogonal projection operator $\mathcal{P}_{T^\perp}$ with $\mathcal{P}_{T^\perp}(M) = (I - UU^*)M(I - VV^*)$. We further define two operators associated with Bernoulli sampling. Let $\Omega_0$ denote a generic subset of $[n] \times [n]$. We define a corresponding projection operator $\mathcal{P}_{\Omega_0}$ as $\mathcal{P}_{\Omega_0}(M) = \sum_{ij} \mathbb{I}_{\{(i,j) \in \Omega_0\}} \langle M, e_i e_j^* \rangle e_i e_j^*$, where $\mathbb{I}_{\{\cdot\}}$ is the indicator function. If $\Omega_0$ is a random set generated by Bernoulli sampling with $\mathbb{P}((i,j) \in \Omega_0) = t_{ij}$ with $0 < t_{ij} \leq 1$ for all $i, j \in [n]$, we further define a linear operator $\mathcal{R}_{\Omega_0}$ as $\mathcal{R}_{\Omega_0}(M) = \sum_{ij} \frac{1}{t_{ij}} \mathbb{I}_{\{(i,j) \in \Omega_0\}} \langle M, e_i e_j^* \rangle e_i e_j^*$.

We further note that throughout this paper "with high probability" means "with probability at least $1 - cn^{-10}$", where the constant $c$ may be different in various contexts.

Our proof includes two main steps: establishing that existence of a certain dual certificate is sufficient to guarantee correct recovery and constructing such a dual certificate. For the first step, we establish the following proposition.

**Proposition 1.** *If* $1 - \rho_{ij} \geq \max \left\{ C_0 \sqrt{\frac{\mu_{ij} r}{n}} \log n, \frac{1}{n^3} \right\}$, *PCP yields a unique solution which agrees with the correct* $(L, S)$ *with high probability if there exists a dual certificate* $Y$ *obeying*

$$\mathcal{P}_\Omega Y = 0, \tag{9}$$

$$\|Y\|_\infty \leq \frac{\lambda}{4}, \tag{10}$$

$$\|\mathcal{P}_{T^\perp}(\lambda \operatorname{sgn}(S) + Y)\| \leq \frac{1}{4}, \tag{11}$$

$$\|\mathcal{P}_T(Y + \lambda \operatorname{sgn}(S) - UV^*)\|_F \leq \frac{\lambda}{n^2} \tag{12}$$

*where* $\lambda = \frac{1}{32\sqrt{n \log n}}$.

The proof of the above proposition adapts the idea in [1,12] for uniform errors to non-uniform errors. In particular, the proof exploits the properties of $\mathcal{R}_\Omega$ associated with non-uniform errors, which are presented as Lemma 1 (established in [9]) and Lemma 2 in Appendix A.1.

Proposition 1 suggests that it suffices to prove Theorem 1 if we find a dual certificate $Y$ that satisfies the dual certificate conditions (9)-(12). Thus, the second step is to construct $Y$ via the golfing scheme. Although we adapt the steps in [12] to construct the dual certificate $Y$, our analysis requires new technical development based on local incoherence parameters. Recall the following definitions in Section 2.1: $\mathbb{P}((i,j) \in \Omega) = \rho_{ij}$ and $\mathbb{P}((i,j) \in \Gamma) = p_{ij}$, where $\Gamma = \Omega^c$ and $p_{ij} = 1 - \rho_{ij}$.

Consider the golfing scheme with nonuniform sizes as suggested in [12] to establish bounds with fewer log factors. Let $\Gamma = \Gamma_1 \cup \Gamma_2 \cup \cdots \cup \Gamma_l$, where $\{\Gamma_k\}$ are independent random sets given by

$$\mathbb{P}((i,j) \in \Gamma_1) = \frac{p_{ij}}{6}, \qquad \mathbb{P}((i,j) \in \Gamma_k) = q_{ij}, \quad \text{for} \quad k = 2, \cdots, l.$$

Thus, if $\rho_{ij} = (1 - \frac{p_{ij}}{6})(1 - q_{ij})^{l-1}$, the two sampling strategies are equivalent. Due to the overlap between $\{\Gamma_k\}$, we have $q_{ij} \geq \frac{5}{6} \frac{p_{ij}}{l-1}$. We set $l = \lfloor 5 \log n + 1 \rfloor$ and construct a dual certificate $Y$ in the following iterative way:

$$Z_0 = \mathcal{P}_T(UV^* - \lambda \operatorname{sgn}(S)) \tag{13}$$
$$Z_k = (\mathcal{P}_T - \mathcal{P}_T \mathcal{R}_{\Gamma_k} \mathcal{P}_T) Z_{k-1}, \quad \text{for} \quad k = 1, \cdots, l \tag{14}$$

$$Y = \sum_{k=1}^{l} \mathcal{R}_{\Gamma_k} Z_{k-1}. \tag{15}$$

It is then sufficient to show that such constructed $Y$ satisfies the dual certificate conditions (9)-(12). Condition (9) is due to the construction of $Y$. Condition (12) can be shown by a concentration property of each iteration step (14) with $\|\cdot\|_F$ characterized in Lemma 3 in Appendix A.1. In order to show that $Y$ satisfies conditions (10) and (11), we introduce the following weighted norm. Let $\hat{w}_{ij} = \sqrt{\frac{\mu_{ij} r}{n^2}}$ and $w_{ij} = \max\{\hat{w}_{ij}, \epsilon\}$, where $\epsilon$ is the smallest nonzero $\hat{w}_{ij}$. Here $\epsilon$ is introduced to avoid singularity. Then for any matrix $Z$, define

$$\|Z\|_{w(\infty)} = \max_{i,j} \frac{|Z_{ij}|}{w_{ij}}. \tag{16}$$

It is easy to verify $\|\cdot\|_{w(\infty)}$ is a well defined norm. We can then show that each iteration step (14) with $\|\cdot\|$ and $\|\cdot\|_{w(\infty)}$ norms satisfies two concentration properties characterized respectively in Lemmas 4 and 5, which are essential to prove conditions (10) and (11).

## 3 Numerical Experiments

In this section, we provide numerical experiments to demonstrate our theoretical results. In these experiments, we adopt an augmented Lagrange multiplier algorithm in [17] to solve the PCP. We set $\lambda = 1/\sqrt{n \log n}$. A trial of PCP (for a given realization of error locations) is declared to be successful if $\hat{L}$ recovered by PCP satisfies $\|\hat{L} - L\|_F / \|L\|_F \leq 10^{-3}$.

We apply the following three models to construct the low rank matrix $L$.

- Bernoulli model: $L = XX^*$ where $X$ is $n \times r$ matrix with entries independently taking values $+1/\sqrt{n}$ and $-1/\sqrt{n}$ equally likely.
- Gaussian model: $L = XX^*$, where $X$ is $n \times r$ matrix with entries independently sampled from Gaussian distribution $\mathcal{N}(0, 1/n)$.
- Cluster model: $L$ is a block diagonal matrix with $r$ equal-size blocks containing all '1's.

In order to demonstrate that the local incoherence parameter affects local robustness to error corruptions, we study the following two types of error corruption models.

- Uniform error corruption: $\operatorname{sgn}(S_{ij})$ is generated as (6) with $\rho_{ij} = \rho$ for all $i, j \in [n]$, and $S = \operatorname{sgn}(S)$.
- Adaptive error corruption: $\operatorname{sgn}(S_{ij})$ is generated as (6) with $\rho_{ij} = \rho \frac{n^2 \sqrt{1/\mu_{ij}}}{\sum_{ij} \sqrt{1/\mu_{ij}}}$ for all $i, j \in [n]$, and $S = \operatorname{sgn}(S)$.

It is clear in both cases, the error matrix has the same average error corruption percentage $\rho$, but in adaptive error corruption, the local error corruption probability is adaptive to the local incoherence.

Our first experiment demonstrates that robustness of PCP to error corruption not only depends on the number of errors but also depends on how errors are distributed over the matrix. For all three

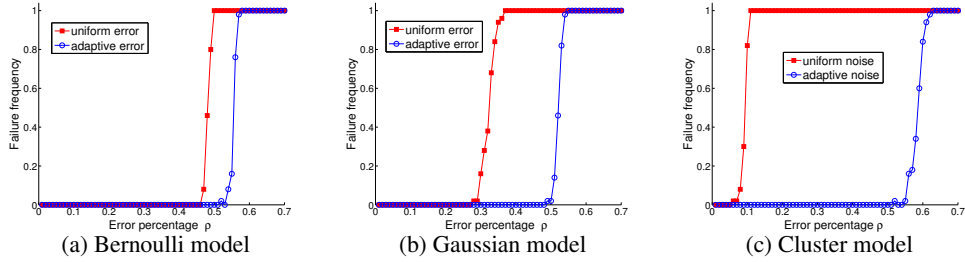

Figure 2: Recovery failure of PCP versus error corruption percentage.

low rank matrix models, we set $n = 1200$ and rank $r = 10$. For each low rank matrix model, we apply the uniform and adaptive error matrices, and plot the failure frequency of PCP versus the error corruption percentage $\rho$ in Fig. 2. For each value of $\rho$, we perform 50 trials of independent error corruption and count the number of failures of PCP. Each plot of Fig. 2 compares robustness of PCP to uniform error corruption (the red square line) and adaptive error corruption (the blue circle line). We observe that PCP can tolerate more errors in the adaptive case. This is because the adaptive error matrix is distributed based on the local incoherence parameter, where error density is higher in areas where matrices can tolerate more errors. Furthermore, comparison among the three plots in Fig. 2 illustrates that the gap between uniform and adaptive error matrices is the smallest for Bernoulli model and the largest for cluster model. Our theoretic results suggest that the gap is due to the variation of the local incoherence parameter across the matrix, which can be measured by the variance of $\mu_{ij}$. Larger variance of $\mu_{ij}$ should yield larger gap. Our numerical calculation of the variances for three models yield $\mathsf{Var}(\mu_{Bernoulli}) = 1.2109$, $\mathsf{Var}(\mu_{Gaussian}) = 2.1678$, and $\mathsf{Var}(\mu_{cluster}) = 7.29$, which confirms our explanation.

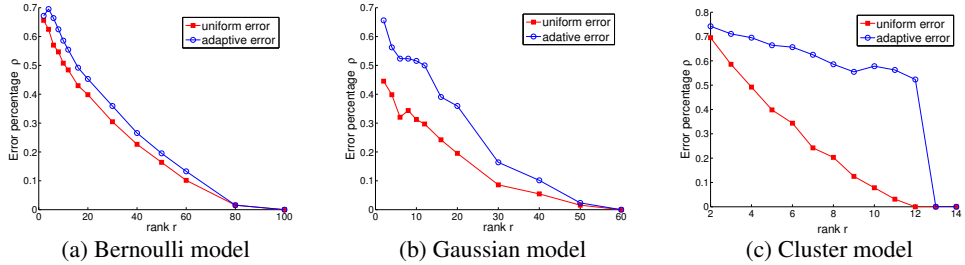

Figure 3: Largest allowable error corruption percentage versus rank of $L$ so that PCP yields correct recovery.

We next study the phase transition in rank and error corruption probability. For the three low-rank matrix models, we set $n = 1200$. In Fig. 3, we plot the error corruption percentage versus the rank of $L$ for both uniform and adaptive error corruption models. Each point on the curve records the maximum allowable error corruption percentage under the corresponding rank such that PCP yields correction recovery. We count a $(r, \rho)$ pair to be successful if nine trials out of ten are successful. We first observe that in each plot of Fig. 3, PCP is more robust in adaptive error corruption due to the same reason explained above. We further observe that the gap between the uniform and adaptive error corruption changes as the rank changes. In the low-rank regime, the gap is largely determined by the variance of incoherence parameter $\mu_{ij}$ as we argued before. As the rank increases, the gap is more dominated by the rank and less affected by the local incoherence. Eventually for large enough rank, no error can be tolerated no matter how errors are distributed.

## 4 Conclusion

We characterize refined conditions under which PCP succeeds to solve the robust PCA problem. Our result shows that the ability of PCP to correctly recover a low-rank matrix from errors is related not only to the total number of corrupted entries but also to locations of corrupted entries, more essentially to the local incoherence of the low rank matrix. Such result is well supported by our numerical experiments. Moreover, our result has rich implication when the low rank matrix is a cluster matrix, and our result coincides with state-of-the-art studies on clustering problems via low rank cluster matrix. Our result may motivate the development of weighted PCP to improve recovery performance similar to the weighted algorithms developed for matrix completion in [9, 18].

## Footnotes

[1] In this paper, we focus on square matrices for simplicity. Our results can be extended to rectangular matrices in a standard way.

[2] $f(n) \in \Theta(g(n))$ means $k_1 \cdot g(n) \leq f(n) \leq k_2 \cdot g(n)$ for some positive $k_1, k_2$.

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
