[Supplementary Material]

# Supplementary Materials

## A  Proof of Theorem 1

### A.1  Key Properties

We provide a number of concentration properties under non-uniform sampling. These properties are in parallel to those under uniform sampling used in [1, 3, 12]. More specifically, Lemma 1 is proven in [9], which readily implies Lemma 3. We develop the proofs for other lemmas based on local incoherence, and provide the detailed proofs in Appendix B.

**Lemma 1.** *[9, Lemma 9] Suppose $\mathbb{P}((i, j) \in \Omega_0) = q_{ij}$ for all $i, j \in [n]$. If $q_{ij} \geq C_0(\mu_{0ij} r \log n)/n$ for some sufficiently large constant $C_0$ and for all $i, j \in [n]$, then with high probability*

$$\|\mathcal{P}_T - \mathcal{P}_T \mathcal{R}_{\Omega_0} \mathcal{P}_T\| \leq \frac{1}{2}. \tag{17}$$

**Lemma 2.** *If $\|\mathcal{P}_T - \mathcal{P}_T \mathcal{R}_{\Omega_0} \mathcal{P}_T\| \leq \frac{1}{2}$ and $p_{ij} \geq p_0$ for all $i, j \in [n]$, then*

  (a)  $\|\mathcal{P}_T \mathcal{R}_{\Omega_0}\| \leq \sqrt{\frac{3}{2p_0}}$;

  (b)  $\mathcal{P}_{\Omega_0} \mathcal{P}_T$ is injective on $T$.

**Lemma 3.** *Suppose $\mathbb{P}((i, j) \in \Omega_0) = q_{ij}$ for all $i, j \in [n]$. For a fixed matrix $Z \in T$, if $q_{ij} \geq C_0(\mu_{ij} r \log n)/n$ for some sufficiently large constant $C_0$ and for all $i, j \in [n]$, then with high probability*

$$\|Z - \mathcal{P}_T \mathcal{R}_{\Omega_0}(Z)\|_F \leq \frac{1}{2}\|Z\|_F. \tag{18}$$

**Lemma 4.** *Suppose $\mathbb{P}((i, j) \in \Omega_0) = q_{ij}$ for all $i, j \in [n]$. For a fixed matrix $Z \in T$, if $q_{ij} \geq C_0 \sqrt{\mu_{ij} r / n}$ for some sufficiently large constant $C_0$ and for all $i, j \in [n]$, then with high probability*

$$\|(\mathcal{R}_{\Omega_0} - I)Z\| \leq \frac{C}{C_0}\|Z\|_{w(\infty)} \tag{19}$$

*for some constant $C$.*

**Lemma 5.** *Suppose $\mathbb{P}((i, j) \in \Omega_0) = q_{ij}$ for all $i, j \in [n]$. Suppose $\beta > 0$ is a scaling factor. For a fixed matrix $Z \in T$, if $q_{ij} \geq C_0 \beta^{-2} \sqrt{\mu_{ij} r / n}$ for some sufficiently large $C_0$ and for all $i, j \in [n]$, then with high probability*

$$\|(\mathcal{P}_T \mathcal{R}_{\Omega_0} - \mathcal{P}_T)Z\|_{w(\infty)} \leq \frac{1}{2}\beta\|Z\|_{w(\infty)}. \tag{20}$$

**Lemma 6.** *Suppose $S$ is the error matrix in the random sign model defined in Section 2.1. Then for any given index $(a, b)$ with $a, b \in [n]$, with high probability*

$$\left|[\mathcal{P}_T \operatorname{sgn}(S)]_{ab}\right| \leq C\sqrt{\frac{\mu_{ab} r \log n}{n}} \tag{21}$$

*for some constant $C$.*

### A.2  Proof of Proposition 1 (Dual Certificate Conditions)

Due to the assumption of the proposition, $\Gamma = \Omega^c$ satisfies the conditions required in Lemma 1. Hence, due to Lemmas 1 and 2, we have $\|\mathcal{P}_T \mathcal{R}_{\Gamma}\| \leq \sqrt{\frac{3}{2p_0}}$ with $p_0 = 1/n^3$ and $\mathcal{P}_{\Gamma} \mathcal{P}_T$ is injective on $T$ with high probability.

Suppose $\hat{L} = L + H$ and $\hat{S} = S - H$ satisfy

$$\|L + H\|_* + \lambda\|S - H\|_1 \leq \|L\|_* + \lambda\|S\|_1. \tag{22}$$

By the definition of subgradient, we have

$$\|L + H\|_* \geq \|L\|_* + \langle \mathcal{P}_T H, UV^* \rangle + \|\mathcal{P}_{T^\perp} H\|_*$$

where we use the fact that there exists $W \in T^\perp$ and $\|W\| \leq 1$ such that $\|\mathcal{P}_{T^\perp} H\|_* = \langle \mathcal{P}_{T^\perp} H, W \rangle$.

Thus, we have

$$\langle \mathcal{P}_T H, UV^* \rangle + \|\mathcal{P}_{T^\perp} H\|_* \leq \lambda\|S\|_1 - \lambda\|S - H\|_1.$$

Furthermore,

$$\|S - H\|_1 = \|S - \mathcal{P}_\Omega H\|_1 + \|\mathcal{P}_\Gamma H\|_1$$
$$\geq \|S\|_1 + \langle \operatorname{sgn}(S), -H \rangle + \|\mathcal{P}_\Gamma H\|_1.$$

Combining the last two inequalities, we have

$$\|\mathcal{P}_{T^\perp} H\|_* + \lambda\|\mathcal{P}_\Gamma H\|_1 \leq \langle H, \lambda \operatorname{sgn}(S) - UV^* \rangle.$$

For a matrix $Y$ that obeys the conditions in the Proposition 1, we derive

$$\langle H, \lambda \operatorname{sgn}(S) - UV^* \rangle$$
$$= \langle H, Y + \lambda \operatorname{sgn}(S) - UV^* \rangle - \langle H, Y \rangle$$
$$= \langle \mathcal{P}_T H, \mathcal{P}_T(Y + \lambda \operatorname{sgn}(S) - UV^*) \rangle + \langle \mathcal{P}_{T^\perp} H, \mathcal{P}_{T^\perp}(Y + \lambda \operatorname{sgn}(S)) \rangle$$
$$\quad - \langle \mathcal{P}_\Gamma H, \mathcal{P}_\Gamma Y \rangle - \langle \mathcal{P}_\Omega H, \mathcal{P}_\Omega Y \rangle$$
$$\leq \frac{\lambda}{n^2}\|\mathcal{P}_T H\|_F + \frac{1}{4}\|\mathcal{P}_{T^\perp} H\|_* + \frac{\lambda}{4}\|\mathcal{P}_\Gamma H\|_1.$$

Combining the previous two inequalities, we obtain

$$\frac{3}{4}\|\mathcal{P}_{T^\perp} H\|_* + \frac{3}{4}\lambda\|\mathcal{P}_\Gamma H\|_1 \leq \frac{\lambda}{n^2}\|\mathcal{P}_T H\|_F.$$

We next bound $\|\mathcal{P}_T H\|_F$ as follows:

$$\|\mathcal{P}_T H\|_F \leq 2\|\mathcal{P}_T \mathcal{R}_\Gamma \mathcal{P}_T(H)\|_F$$
$$\leq 2\|\mathcal{P}_T \mathcal{R}_\Gamma \mathcal{P}_{T^\perp}(H)\|_F + 2\|\mathcal{P}_T \mathcal{R}_\Gamma(H)\|_F$$
$$\leq \sqrt{\frac{6}{p_0}}\|\mathcal{P}_{T^\perp}(H)\|_F + \sqrt{\frac{6}{p_0}}\|\mathcal{P}_\Gamma(H)\|_F.$$

We thus obtain

$$\left(\frac{3}{4} - \frac{\lambda}{n^2}\sqrt{\frac{6}{p_0}}\right)\|\mathcal{P}_{T^\perp}(H)\|_F + \left(\frac{3}{4}\lambda - \frac{\lambda}{n^2}\sqrt{\frac{6}{p_0}}\right)\|\mathcal{P}_\Gamma(H)\|_F \leq 0.$$

The above inequality implies that if $p_0 \geq 1/n^3$, then $\mathcal{P}_{T^\perp} H = \mathcal{P}_\Gamma H = 0$. This further implies $\mathcal{P}_\Gamma \mathcal{P}_T(H) = 0$. Since $\mathcal{P}_\Gamma \mathcal{P}_T$ is injective on $T$, we have $\mathcal{P}_T H = 0$. Consequently, $H = 0$.

## A.3   Dual Certificate Verification

We show that the dual certificate constructed in (13)-(15) satisfies the conditions in Proposition 1.

We first bound $\|Z_0\|_F, \|Z_0\|_\infty$ and $\|Z_0\|_{w(\infty)}$. Observe that for an index pair $(a, b)$, we have

$$|[Z_0]_{ab}| \leq |[UV^*]_{ab}| + \lambda|[\mathcal{P}_T \operatorname{sgn}(S)]_{ab}|.$$

Using the fact that $|[UV^*]_{ab}| \leq \sqrt{\frac{\mu_{ab} r}{n^2}}$ and $\lambda = \frac{1}{32\sqrt{n \log n}}$, and applying Lemma 6, we obtain

$$\|Z_0\|_\infty \leq C\sqrt{\mu r}/n. \tag{23}$$

Furthermore,

$$\|Z_0\|_F \leq \|UV^*\|_F + \lambda\|\mathcal{P}_T \operatorname{sgn}(S)\|_F \leq \sqrt{r} + C\sqrt{\mu r} \leq C'\sqrt{\mu r} \tag{24}$$

where we used $\|Z\|_F \leq n\|Z\|_\infty$ for any matrix $Z$, and

$$\|Z_0\|_{w(\infty)} \leq \|UV^*\|_{w(\infty)} + \lambda\|\mathcal{P}_T \operatorname{sgn}(S)\|_{w(\infty)}$$
$$\leq 1 + \max_{a,b} \lambda\frac{|[\mathcal{P}_T \operatorname{sgn}(S)]_{ab}|}{w_{ab}}$$
$$\leq C', \tag{25}$$

where we used the definition $w_{ab} = \max\{\sqrt{\mu_{ab} r/n^2}, \epsilon\}$ and $\lambda = \frac{1}{32\sqrt{n \log n}}$. We note that for the sake of convenience, the constants $C$ and $C'$ may be different from line to line.

We further note that Lemma 3 implies

$$\|Z_k\|_F \leq \frac{1}{2}\|Z_{k-1}\|_F \tag{26}$$

with high probability, provided that $q_{ij} \geq C_0(\mu_{ij}r \log n)/n$ for some sufficiently large constant $C_0$ and for all $i, j \in [n]$.

Lemma 4 implies

$$\|(I - \mathcal{R}_{\Gamma_k})Z_{k-1}\| \leq \frac{C}{C_0}\|Z_{k-1}\|_{w(\infty)} \tag{27}$$

with high probability, provided that $q_{ij} \geq C_0\sqrt{\mu_{ij}r/n}$ for some sufficiently large constant $C_0$ and for all $i, j \in [n]$.

Lemma 5 implies

$$\|Z_1\|_{w(\infty)} \leq \frac{1}{2\sqrt{\log n}}\|Z_0\|_{w(\infty)} \tag{28}$$

and

$$\|Z_k\|_{w(\infty)} \leq \frac{1}{2}\|Z_{k-1}\|_{w(\infty)} \quad \text{for } k = 2, \cdots, l \tag{29}$$

with high probability, provided that $q_{ij} \geq C_0\sqrt{\mu_{ij}r/n}$ for some sufficiently large constant $C_0$ and for all $i, j \in [n]$.

We are now ready to show that the constructed dual certificate $Y$ obeys the conditions (9)-(12) in Proposition 1. Clearly, $Y$ satisfies $\mathcal{P}_\Omega Y = 0$ given in (9) due to the construction.

In order to show that $Y$ satisfies (12), we derive

$$\|\mathcal{P}_T Y + \mathcal{P}_T(\lambda \operatorname{sgn}(S) - UV^*)\|_F$$

$$= \left\| Z_0 - \left( \sum_{k=1}^{l} \mathcal{P}_T \mathcal{R}_{\Gamma_k} Z_{k-1} \right) \right\|_F$$

$$= \left\| (\mathcal{P}_T - \mathcal{P}_T \mathcal{R}_{\Gamma_1})Z_0 - \left( \sum_{k=2}^{l} \mathcal{P}_T \mathcal{R}_{\Gamma_k} Z_{k-1} \right) \right\|_F$$

$$= \left\| \mathcal{P}_T Z_1 - \left( \sum_{k=1}^{l} \mathcal{P}_T \mathcal{R}_{\Gamma_k} Z_{k-1} \right) \right\|_F$$

$$= \cdots$$

$$= \|Z_l\|_F \overset{(a)}{\leq} \left(\frac{1}{2}\right)^l \cdot \|Z_0\|_F \overset{(b)}{\leq} C'\left(\frac{1}{2}\right)^l \sqrt{\mu r} \leq \frac{\lambda}{n^2},$$

where (a) follows from (26) and (b) follows from (24).

In order to show that $Y$ satisfies (11), we respectively show that $\|\mathcal{P}_{T^\perp} Y\| \leq \frac{1}{8}$ and $\|\mathcal{P}_{T^\perp}(\lambda \operatorname{sgn}(S))\| \leq \frac{1}{8}$ as follows.

$$\|\mathcal{P}_{T^\perp} Y\| = \left\| \mathcal{P}_{T^\perp} \sum_{k=1}^{l} \mathcal{R}_{\Gamma_k} Z_{k-1} \right\|$$

$$\leq \sum_{k=1}^{l} \|\mathcal{P}_{T^\perp} \mathcal{R}_{\Gamma_k} Z_{k-1}\|$$

$$\overset{(a)}{=} \sum_{k=1}^{l} \|\mathcal{P}_{T^\perp}(\mathcal{R}_{\Gamma_k} Z_{k-1} - Z_{k-1})\|$$

$$\leq \sum_{k=1}^{l} \|\mathcal{R}_{\Gamma_k} Z_{k-1} - Z_{k-1}\|$$

$$\overset{(b)}{\leq} \sum_{k=1}^{l} \frac{C}{C_0}\|Z_{k-1}\|_{w(\infty)}$$

$$\overset{(c)}{\leq} \frac{C}{C_0}\left(1 + \sum_{k=2}^{l} \frac{1}{\sqrt{\log n}}\left(\frac{1}{2}\right)^{k-1}\right)\|Z_0\|_{w(\infty)}$$

$$\leq \frac{2C}{C_0}\|Z_0\|_{w(\infty)}$$

$$\overset{(d)}{\leq} \frac{1}{8},$$

where (a) follows because $Z_{k-1} \in T$, (b) follows from (27), (c) follows from (28) and (29), and (d) follows from (25) and $C_0$ is sufficiently large.

Furthermore, by applying the spectral norm bound on random matrix in [19], we have

$$\|\mathcal{P}_{T^\perp}(\lambda \operatorname{sgn}(S))\| \leq \lambda \|\operatorname{sgn}(S)\| \leq \lambda \cdot 4\sqrt{n}. \tag{30}$$

Since $\lambda = \frac{1}{32\sqrt{n \log n}}$, we have

$$\|\mathcal{P}_{T^\perp}(\lambda \operatorname{sgn}(S))\| \leq \frac{1}{8\sqrt{\log n}} \leq \frac{1}{8}.$$

In order to show that $Y$ satisfies (10), we derive

$$
\begin{aligned}
\|Y\|_\infty &= \left\| \sum_{k=1}^l \mathcal{R}_{\Gamma_k} Z_{k-1} \right\|_\infty \\
&\stackrel{(a)}{\leq} \left\| \sum_{i,j} \frac{6}{p_{ij}} \mathbb{I}_{\{(i,j) \in \Gamma_1\}} (Z_0)_{ij} e_i e_j^* \right\|_\infty + \sum_{k=2}^l \left\| \sum_{i,j} \frac{1}{q_{ij}} \mathbb{I}_{\{(i,j) \in \Gamma_k\}} (Z_{k-1})_{ij} e_i e_j^* \right\|_\infty \\
&\leq 6 \cdot \max_{i,j} \frac{|(Z_0)_{ij}|}{p_{ij}} + \sum_{k=2}^l \max_{i,j} \frac{|(Z_{k-1})_{ij}|}{q_{ij}} \\
&\leq \frac{6}{C_0 \sqrt{n} \log n} \|Z_0\|_{w(\infty)} + \sum_{k=2}^l \frac{1}{C_0 \sqrt{n}} \|Z_{k-1}\|_{w(\infty)} \\
&\stackrel{(b)}{\leq} \frac{6}{C_0 \sqrt{n} \log n} \|Z_0\|_{w(\infty)} + \sum_{k=2}^l \frac{1}{C_0 \sqrt{n} \log n} \left(\frac{1}{2}\right)^{k-1} \|Z_0\|_{w(\infty)} \\
&\leq \frac{7}{C_0 \sqrt{n} \log n} \|Z_0\|_{w(\infty)} \\
&\stackrel{(c)}{\leq} \frac{224 C}{C_0} \lambda \\
&\stackrel{(d)}{\leq} \frac{\lambda}{4},
\end{aligned}
$$

where (a) is due to the golfing scheme with non-uniform partitions, (b) follows from (28) and (29), (c) follows from (25) and (d) follows because $C_0$ is sufficiently large.

# B  Proofs of Key Properties

In this section, we prove the key lemmas provided in Appendix A.1. The central technique used here is non-communicative Bernstein inequality [20].

## B.1  Proof of Lemma 2

We note that the condition $\|\mathcal{P}_T - \mathcal{P}_T \mathcal{R}_{\Omega_0} \mathcal{P}_T\| \leq \frac{1}{2}$ implies for any matrix $Z$

$$\frac{1}{2}\|\mathcal{P}_T Z\|_F \leq \|\mathcal{P}_T \mathcal{R}_{\Omega_0} \mathcal{P}_T(Z)\|_F \leq \frac{3}{2}\|\mathcal{P}_T Z\|_F.$$

Thus, for any matrix $Z$, we have

$$
\begin{aligned}
\left\| \mathcal{R}_{\Omega_0}^{1/2} \mathcal{P}_T(Z) \right\|_F^2 &= \langle \mathcal{R}_{\Omega_0}^{1/2} \mathcal{P}_T(Z), \mathcal{R}_{\Omega_0}^{1/2} \mathcal{P}_T(Z) \rangle \\
&= \langle Z, (\mathcal{R}_{\Omega_0}^{1/2} \mathcal{P}_T)^* \mathcal{R}_{\Omega_0}^{1/2} \mathcal{P}_T(Z) \rangle \\
&= \langle \mathcal{P}_T(Z), \mathcal{P}_T \mathcal{R}_{\Omega_0} \mathcal{P}_T(Z) \rangle \\
&\leq \|\mathcal{P}_T Z\|_F \|\mathcal{P}_T \mathcal{R}_{\Omega_0} \mathcal{P}_T(Z)\|_F \\
&\leq \frac{3}{2} \|\mathcal{P}_T Z\|_F^2.
\end{aligned}
$$

Thus, $\left\| \mathcal{R}_{\Omega_0}^{1/2} \mathcal{P}_T \right\| \leq \sqrt{3/2}$ and hence $\left\| \mathcal{P}_T \mathcal{R}_{\Omega_0}^{1/2} \right\| \leq \sqrt{3/2}$ because $\mathcal{R}_{\Omega_0}^{1/2} \mathcal{P}_T$ and $\mathcal{P}_T \mathcal{R}_{\Omega_0}^{1/2}$ are adjoint operators and have equal norm. On the other hand, we show $\left\| \mathcal{R}_{\Omega_0}^{1/2} \right\| \leq 1/\sqrt{p_0}$ as follows. For any matrix

$Z$,

$$\left\| \mathcal{R}_{\Omega_0}^{1/2}(Z) \right\|_F^2 = \left\| \sum_{i,j} \frac{1}{\sqrt{p_{ij}}} \mathbb{I}_{\{(i,j) \in \Omega_0\}} Z_{ij} e_i e_j^* \right\|_F^2$$

$$\leq \sum_{i,j} \frac{Z_{ij}^2}{p_{ij}} \leq \frac{1}{p_0} \|Z\|_F^2.$$

Thus, $\|\mathcal{R}_{\Omega_0} \mathcal{P}_T\| \leq \|\mathcal{R}_{\Omega_0}^{1/2}\| \cdot \|\mathcal{R}_{\Omega_0}^{1/2} \mathcal{P}_T\| \leq \sqrt{\frac{3}{2p_0}}$. Thus, $\|\mathcal{P}_T \mathcal{R}_{\Omega_0}\| \leq \sqrt{\frac{3}{2p_0}}$.

Since we have $\frac{1}{2}\|\mathcal{P}_T Z\|_F \leq \|\mathcal{P}_T \mathcal{R}_{\Omega_0} \mathcal{P}_T(Z)\|_F \leq \frac{3}{2}\|\mathcal{P}_T Z\|_F$ for any matrix $Z \in T$, the operator $\mathcal{P}_T \mathcal{R}_{\Omega_0} \mathcal{P}_T$ mapping $T$ onto itself is well conditioned. Thus, $\mathcal{P}_{\Omega_0} \mathcal{P}_T$ is injective on $T$, i.e., for $Z \in T$, $\mathcal{P}_{\Omega_0} \mathcal{P}_T(Z) = 0$ if and only if $Z = 0$.

## B.2 Proof of Lemma 4

Let $\delta_{ij}$ denote the Bernoulli random variable $\mathbb{I}((i,j) \in \Omega_0)$. We can derive

$$(\mathcal{R}_{\Omega_0} - I)Z = \sum_{i,j} \left( \frac{1}{q_{ij}} \delta_{ij} - 1 \right) \langle e_i e_j^*, Z \rangle e_i e_j^*$$

$$=: \sum_{i,j} X_{ij}.$$

We note that $X_{ij}$ for all $i, j \in [n]$ are zero-mean independent random matrices. Furthermore,

$$\|X_{ij}\| \leq \frac{1}{q_{ij}} |Z_{ij}| \leq \frac{1}{C_0 \sqrt{n}} \|Z\|_{w(\infty)}.$$

and

$$\left\| \sum_{i,j} \mathbb{E} X_{ij} X_{ij}^* \right\| = \left\| \sum_{i,j} \mathbb{E} \left( \frac{1}{q_{ij}} \delta_{ij} - 1 \right)^2 Z_{ij}^2 e_i e_i^* \right\|$$

$$= \left\| \sum_{i,j} \left( \frac{1}{q_{ij}} - 1 \right) Z_{ij}^2 e_i e_i^* \right\|$$

$$\leq \max_i \sum_j \frac{Z_{ij}^2}{q_{ij}}$$

$$\leq n \|Z\|_{w(\infty)}^2 \cdot \max_{i,j} \frac{w_{ij}^2}{q_{ij}}$$

$$\leq \|Z\|_{w(\infty)}^2 \cdot \frac{1}{C_0^2} \max_{i,j} \left( C_0 \sqrt{\frac{\mu_{ij} r}{n}} \right)$$

$$\leq \frac{1}{C_0^2 \log n} \|Z\|_{w(\infty)}^2$$

Similarly, it can be shown that $\|\sum_{i,j} \mathbb{E} X_{ij}^* X_{ij}\| \leq \frac{1}{C_0^2 \log n} \|Z\|_{w(\infty)}^2$. Thus, applying the non-commutative Bernstein inequality, we obtain

$$\|(\mathcal{R}_{\Omega_0} - I)Z\| = \left\| \sum_{i,j} X_{ij} \right\|$$

$$\leq C \left( \sqrt{\frac{1}{C_0^2 \log n} \|Z\|_{w(\infty)}^2 \cdot \log n} + \frac{1}{C_0 \sqrt{n}} \|Z\|_{w(\infty)} \cdot \log n \right)$$

$$\leq \frac{C}{C_0} \|Z\|_{w(\infty)}$$

with high probability.

## B.3 Proof of Lemma 5

For any entry index pair $(a, b)$, we have

$$[(\mathcal{P}_T \mathcal{R}_{\Omega_0} - \mathcal{P}_T)Z]_{ab} \cdot \frac{1}{w_{ab}}$$

$$= \sum_{i,j} \left( \frac{1}{q_{ij}} \delta_{ij} - 1 \right) Z_{ij} \langle \mathcal{P}_T(e_i e_j^*), e_a e_b^* \rangle \cdot \frac{1}{w_{ab}}$$

$$= \sum_{i,j} \left( \frac{1}{q_{ij}} \delta_{ij} - 1 \right) Z_{ij} \langle e_i e_j^*, \mathcal{P}_T(e_a e_b^*) \rangle \cdot \frac{1}{w_{ab}}$$

$$=: \sum_{i,j} x_{ij}.$$

We note that $x_{ij}$ for $i, j \in [n]$ are independent random variables and $\mathbb{E} x_{ij} = 0$. Furthermore,

$$|x_{ij}| \le \frac{1}{q_{ij}} |Z_{ij}| \cdot |\langle e_i e_j^*, \mathcal{P}_T(e_a e_b^*) \rangle| \cdot \frac{1}{w_{ab}}$$

$$\le |Z_{ij}| \cdot \frac{1}{C_0 \beta^{-2} \sqrt{\mu_{ij} r / n}} \cdot \sqrt{\frac{2\mu_{ij} r}{n}} \cdot \sqrt{\frac{2\mu_{ab} r}{n}} \cdot \frac{1}{\sqrt{\frac{\mu_{ab} r}{n^2}}}$$

$$\le \frac{2\beta^2}{C_0} \sqrt{\frac{\mu r}{n}} \frac{|Z_{ij}|}{w_{ij}}$$

$$\le \frac{2\beta^2}{C_0^2 \log n} \|Z\|_{w_{(\infty)}},$$

and

$$\left| \sum_{i,j} \mathbb{E} x_{ij}^2 \right| \le \sum_{i,j} \mathbb{E} \left( \frac{1}{q_{ij}} \delta_{ij} - 1 \right)^2 Z_{ij}^2 \cdot |\langle e_i e_j^*, \mathcal{P}_T(e_a e_b^*) \rangle|^2 \cdot \frac{1}{w_{ab}^2}$$

$$\le \sum_{i,j} \left( \frac{1}{q_{ij}} - 1 \right) \frac{Z_{ij}^2}{w_{ij}^2} \cdot \frac{w_{ij}^2}{w_{ab}^2} \cdot |\langle e_i e_j^*, \mathcal{P}_T(e_a e_b^*) \rangle|^2$$

$$\le \frac{1}{C_0 \beta^{-2}} \sqrt{n \mu r} \cdot \|Z\|_{w_{(\infty)}}^2 \cdot \frac{1}{\mu_{ab} r} \|\mathcal{P}_T(e_a e_b^*)\|_F^2$$

$$\le \frac{2\beta^2}{C_0} \sqrt{\frac{\mu r}{n}} \cdot \|Z\|_{w_{(\infty)}}^2$$

$$\le \frac{2\beta^2}{C_0^2 \log n} \|Z\|_{w_{(\infty)}}^2,$$

where we use the fact $\|\mathcal{P}_T(e_a e_b^*)\|_F^2 \le \frac{2\mu_{ab} r}{n}$, and the last steps of the above two derivations are due to the fact $C_0 \sqrt{\mu r / n} \log n \le 1$ implied by our assumption.

Thus, applying the non-commutative Bernstein inequality, we have

$$\left| \sum_{i,j} x_{ij} \right| \le C \left( \sqrt{\frac{2\beta^2}{C_0^2 \log n} \|Z\|_{w_{(\infty)}}^2 \cdot \log n} + \frac{2\beta^2}{C_0^2 \log n} \|Z\|_{w_{(\infty)}} \cdot \log n \right)$$

$$= C \left( \frac{\sqrt{2}}{C_0} \beta + \frac{2}{C_0^2} \beta^2 \right) \|Z\|_{w_{(\infty)}}$$

$$\le \frac{1}{2} \beta \|Z\|_{w_{(\infty)}},$$

with high probability, provided that $C_0$ is sufficiently large.

## B.4 Proof of Lemma 6

We first express $(\mathcal{P}_T \operatorname{sgn}(S))_{ab}$ as

$$\langle e_a e_b^*, \mathcal{P}_T \operatorname{sgn}(S) \rangle = \langle \operatorname{sgn}(S), \mathcal{P}_T(e_a e_b^*) \rangle$$

$$= \sum_{i,j} \delta_{ij} \langle e_i e_j^*, \mathcal{P}_T(e_a e_b^*) \rangle$$

$$=: \sum_{i,j} x_{ij}$$

where

$$\delta_{ij} = \begin{cases} 1 & \text{with prob.} & \rho_{ij}/2 \\ 0 & \text{with prob.} & 1 - \rho_{ij} \\ -1 & \text{with prob.} & \rho_{ij}/2. \end{cases}$$

We note that $x_{ij}$ for $i, j \in [n]$ are independent random variables and $\mathbb{E} x_{ij} = 0$. Furthermore, by applying Cauchy-Schwartz inequality and the fact $\|\mathcal{P}_T(e_a e_b^*)\|_F^2 \leq \frac{2\mu_{ab} r}{n}$, we have

$$|x_{ij}| \leq |\langle e_i e_j^*, \mathcal{P}_T(e_a e_b^*) \rangle| \leq \sqrt{\frac{2\mu r}{n}} \cdot \sqrt{\frac{2\mu_{ab} r}{n}}$$

and

$$\begin{aligned} \left| \sum_{i,j} \mathbb{E} x_{ij}^2 \right| &= \left| \sum_{i,j} \mathbb{E} \delta_{ij}^2 \langle e_i e_j^*, \mathcal{P}_T(e_a e_b^*) \rangle^2 \right| \\ &= \left| \sum_{i,j} \rho_{ij} \langle e_i e_j^*, \mathcal{P}_T(e_a e_b^*) \rangle^2 \right| \\ &\leq \left| \sum_{i,j} \langle e_i e_j^*, \mathcal{P}_T(e_a e_b^*) \rangle^2 \right| \\ &= \|\mathcal{P}_T(e_a e_b^*)\|_F^2 \\ &\leq \frac{2\mu_{ab} r}{n}. \end{aligned}$$

Thus, applying the non-commutative Bernstein inequality, we obtain

$$\begin{aligned} \left| \sum_{i,j} x_{ij} \right| &\leq C \left( \sqrt{\frac{2\mu_{ab} r}{n} \cdot \log n} + \sqrt{\frac{2\mu r}{n}} \cdot \sqrt{\frac{2\mu_{ab} r}{n}} \cdot \log n \right) \\ &\leq C \sqrt{\frac{\mu_{ab} r \log n}{n}}, \end{aligned}$$

where the last inequality follows from the fact $C_0 \sqrt{\mu r / n} \log n \leq 1$ implied by the assumption.