[Reviews · NeurIPS 2015]

Submitted by Assigned_Reviewer_1

The paper presents an analysis of the robust PCA problem (decompose M=L+S) but with random locations of the locations of the errors S and a local notion of incoherence.

That is rather than a global incoherence of L, authors study local incoherence of each element of L. Using this notion, they derive tighter bounds for the number of non-zeros allowed in S.

The result of the paper seems novel and somewhat interesting. The analysis follows the standard analysis but includes complications due to the local incoherence. However, it is not clear if the local incoherence indeed adds a lot of complication to the proof. Also, the main message or technique from the analysis is not concrete.

So, despite the result being novel, it is not clear how interesting the analysis is and how important the result is.
Summary: See below.

Submitted by Assigned_Reviewer_2

Summary of paper:

Until recently, the related but different problems of matrix completion and robust PCA - via nuclear norm minimization - had similar assumptions under which recovery was known: incoherence of the low-rank matrix, and uniform randomness in the samples (for completion) or errors (for robust PCA).

Recently however it was shown that for matrix completion recovery is provably possible, via same convex methods, even when the sampling is non-uniform, provided this non-uniformity is adapted to LOCAL coherence.

This paper establishes a similar result for robust PCA, showing that if the errors are non-uniromly random (again in a way that is adapted to the same local coherences) then exact recovery is again possible.

Quality:

Overall the result is likely to be important and of interest. The techniques are new, but along lines somewhat similar to those employed in the (recent) matrix completion analysis using local coherence.

The experiments are not very convincing because the low-rank matrices are constructed to have very similar local coherences. It would have been better to see matrices with very different local coherences. This can be done by pre and post multiplying the low-rank matrices generated in the paper with diagonal matrices that have large dynamic range on the diagonal elements.

Clarity:

The paper is clearly written, though by the very nature of what it is trying to do is a bit dense to read.

Originality:

The paper takes the new recent understanding of how local coherence can allow for non-uniformity in sampling for matrix completion, and extends it to the case of errors in robust PCA. The results are original.

Significance:

Robust PCA has a huge number of applications, and by this token an improvement in its understanding is significant. However the paper does not provide any new method; but rather a better analysis of the already popular method. So its empirical significance may be lower.

It would have been nice to see specific examples where this kind of non-uniformity both naturally arises, and leads to appreciably better recovery. For example, does this imply that for the graph clustering problem with unequal size clusters, larger clusters can be sparser but still recover-able ?
Summary: The paper proves interesting results on the ability of the standard robust PCA algorithm to recover from non-uniform errors, provided these are adapted to the local coherences of the matrix. The analysis follows along lines of recent work, but has a couple interesting innovations.

Submitted by Assigned_Reviewer_3

This paper considers the Robust PCA problem ("low-rank plus sparse") where the leverage scores (or local incoherence) of the low-rank matrix and the non-zero probabilities of the sparse matrix are allowed to vary across the entries. Sufficient conditions for exact recovery are provided in terms of local relations of the leverage scores and error probabilities.

This paper is in line with previous work on matrix completion under a similar non-uniform setting. The analysis uses the techniques of golfing schemes and weighted norms, which are developed in previous work.

My main concern is that applications of such a result are not immediately clear. In particular, what is a scenario where the error probabilities and leverage scores will happen to align with each other and satisfy the conditions in the paper? Unlike matrix completion where the observation probabilities might be controllable in certain cases, here the error probabilities are not. More discussion would help.

The clustering problem mentioned in the paper could be a potential application. As the local incoherence is related to the cluster sizes, perhaps larger clusters are allowed to have higher error probabilities? It would be useful to write down a corollary for this problem.

Other comments:

1) It would be helpful to formally write down the elimination/derandomization arguments for this non-uniform setting.

2) The results involve both \mu_0 and \mu_1. It would help to explicitly point out in which part of the proof is \mu_1 needed.

Summary: This paper extends previous work on matrix completion with non-uniform local incoherence. The theoretical results appear correct, and the proof is based on previously developed techniques. It would be good if the authors can better motivate the problem setting and discuss applications.

Submitted by Assigned_Reviewer_4

The authors provide an analysis of the Robust PCA problem when the sparse corruption follow a non-uniform Bernoulli distribution. The paper shows that in such a non-uniform situation, high probability recovery relies on local incoherence of the low-rank matrix. In particular, entries which are locally incoherent can tolerate more error.

The analysis builds on certain existing ideas in the literature, such as the golfing scheme. The results are presented both for the random sign setting as well as the fixed sign setting. The results do provide additional insights into the robust PCA problem.

Some concerns regarding the work - clarifications on these may help the reader better understand the contribution. First, there has been considerable progress in convex demixing [1], which provides general geometric conditions under which recovery of the form S+L is possible - in fact, the scope of these developments are substantially more general. It will be important to contrast the proposed specific results to this body of work. Further, the local incoherence condition, while interesting, does not appear testable for a given problem. So, it is somewhat unclear how to use the condition in practice.

The analysis in the paper is based on weighted norm, which in itself is an interesting idea. But several conclusions concerning this norm can be found in your reference [9]. It will be important to clearly separate what is known, and what the current paper adds, and otherwise highlight the advantage of this norm for the current problem. Otherwise, the contributions come across as somewhat incremental.

The implications for cluster matrices is interesting. The experimental part is fine. It may be interesting to have some results on robust PCA after centering to see if one gets qualitatively different results.

Additional comments -

Pros:

Clear writing

Many experiments

Cons:

Results not put in proper context of existing related literature.

Draws considerably from existing results, and comes across as incremental.

[1] M. B. McCoy, A geometric analysis of convex demixing, 2013.
Summary: The authors provide an analysis of the Robust PCA problem when the sparse corruption follow a non-uniform Bernoulli distribution. The paper shows that in such a non-uniform situation, high probability recovery relies on local incoherence of the low-rank matrix. In particular, entries which are locally incoherent can tolerate more error.

The analysis builds on certain existing ideas in the literature, such as the golfing scheme. The results are presented both for the random sign setting as well as the fixed sign setting. The results do provide additional insights into the robust PCA problem. There are some concerns regarding the work - clarifications on these may help the reader better understand the contribution.

Author Feedback
Author rebuttal: We thank reviewers for their comments. We first summarize our contributions and then provide point-to-point responses.

In contrast to all previous studies of robust PCA that assume uniform noise corruption, we investigate the nontrivial extension under nonuniform noise corruption. We characterize performance guarantee of PCP algorithm by exploiting local incoherence to deal with nonuniform noise. Our result offers crystallized and deeper insights beyond classic Robust PCA, as demonstrated by the new implications on graph clustering. Our technical innovation lies in constructing a novel weighted norm (differently from existing work) and establishing its statistical properties, which prove to be powerful and significantly facilitate successful proof of performance guarantee.

1. Reviewers 1 and 2 comment on applications of our results and suggest clustering problem as an application. They also ask: does the result imply that for graph clustering problem, larger clusters can be sparser but still recoverable? Or equivalently, are larger clusters allowed to have higher error probabilities?

Response: Thanks for pointing out the clustering problem as an interesting application. Indeed, our result implies that larger clusters can be sparser but still recoverable, or equivalently, clusters with larger sizes can allow higher error probabilities. More discussions can be found in Sec. 2.3 of the paper.

2. Reviewer 2 comments that the results involve both \mu_0 and \mu_1. It would help to explicitly point out in which part of the proof is \mu_1 needed.

Response: \mu_1 is naturally involved to prove the key property Lemma 5 associated with the weighted infinity norm (see Suppl Sec. B.3). This property serves a central role for dual certificate verification (see Suppl Sec. A.3).

3. Reviewer 3's comment 1: The analysis in this paper is based on weighted norm, which in itself is an interesting idea. But several conclusions concerning this norm can be found in your reference [9]. It will be important to clearly separate what is known, and what the current paper adds, and otherwise highlight the advantage of this norm for the current problem. Otherwise, the contributions come across as somewhat incremental.

Response: In fact, our new weighted norm is very different from the norms in [9]. In particular, our weighted norm involves both \mu_0 and \mu_1, whereas the norms in [9] involve only \mu_0. Hence, conclusions on the norms in [9] are not applicable here. One major contribution of this paper lies in developing new proofs of statistical properties associated with our new norm (see Suppl Sec. B.2, B.3, B.4). Moreover, applying these key properties to prove performance guarantee further requires considerable technical efforts. Thus, the analysis does contain substantial originality.

4. Review 3's comment 2: There has been considerable progress in convex demixing [McCoy'13]. It will be important to contrast the proposed specific results to this body of work.

Response: Indeed, [McCoy'13] considers the separation of multiple signals in a more general scope. When [McCoy'13] is specialized to robust PCA problem, the technical assumptions and results are substantially different from our work as we describe below.

First, assumptions on matrices in [McCoy'13] are very different from those in our paper. In [McCoy'13], L is randomly generated and S is a sparse matrix randomly rotated. In our paper, L is assumed to be unknown but deterministic, and S has each entry randomly generated by non-uniform Bernoulli distribution.

Second, [McCoy'13] analyzes an algorithm that requires the knowledge of sparsity level of S, whereas our paper studies PCP algorithm that does not require such knowledge.

Thus, different assumptions and algorithms naturally lead to different results and interpretations. While [McCoy'13] focuses on rank-sparsity phase transition when L and S are generally incoherent, our result characterizes how incoherence of L and S affects ability of PCP to recover L and S.

5. Reviewer 5's comment: The analysis follows the standard analysis but includes complications due to local incoherence. However, it is not clear if local incoherence indeed adds a lot of complication to the proof. Also, the main message or technique from the analysis is not concrete. The reviewer further asked "how important the result is".

Response: The major difference of our analysis from the standard proof lies in constructing a new weighted norm for dealing with local incoherence. In fact, proving statistical properties associated with this new norm and further exploiting these properties to prove performance guarantee require considerable technical developments.

To justify the importance, our result for robust PCA with nonuniform noise can yield new and interesting insights beyond what classical robust PCA offers, for example, for graph clustering problem.